# DNA sequence classification for diabetes mellitus using NuSVC and XGBoost: A comparative

**Said A. Salloum**[1]*, **Khaled Mohammad Alomari**[2], **Ayham Salloum**[3]

**1** School of Computing, Skyline University College, Sharjah, UAE, **2** Faculty of Information Technology, Abu Dhabi University, Abu Dhabi, UAE, **3** College of Medicine, University of Sharjah, Sharjah, UAE

* said.salloum@skylineuniversity.ac.ae, dkhaled.alomari@adu.ac.ae, ssalloum1978@gmail.com

## Abstract

Diabetes Mellitus is a global health concern, characterized by high blood sugar levels over a prolonged period, leading to severe complications if left unmanaged. The early identification of individuals at risk is critical for effective intervention and treatment. Traditional diagnostic methods rely heavily on clinical symptoms and biochemical tests, which may not capture the underlying genetic predispositions. With the advent of genomics, DNA sequence analysis has emerged as a promising approach to uncover the genetic markers associated with Diabetes Mellitus. However, the challenge lies in accurately classifying DNA sequences to predict susceptibility to the disease, given the complex nature of genetic data. This study addresses this challenge by employing two advanced machine learning models, NuSVC (Nu-Support Vector Classification) and XGBoost (Extreme Gradient Boosting), to classify DNA sequences related to Diabetes Mellitus. The dataset, obtained from reputable sources like NCBI, was preprocessed using Natural Language Processing (NLP) techniques, where DNA sequences were treated as textual data and transformed into numerical features using TF-IDF (Term Frequency-Inverse Document Frequency). To handle the class imbalance in the dataset, SMOTE (Synthetic Minority Over-sampling Technique) was applied. The models were trained and validated using 10-fold cross-validation. XGBoost was trained with up to 300 boosting rounds, and performance was evaluated using accuracy, precision, recall, F1-score, ROC-AUC, and log loss. The results demonstrate that XGBoost outperformed NuSVC across all metrics, achieving an accuracy of 98%, a log loss of 0.0650, and an AUC of 1.00, compared to NuSVC's accuracy of 87%, log loss of 0.2649, and AUC of 0.95. The superior performance of XGBoost indicates its robustness in handling complex genetic data and its potential utility in clinical applications for early diagnosis of Diabetes Mellitus. The findings of this study underscore the importance of advanced machine learning techniques in genomics and suggest that integrating such models into healthcare systems could significantly enhance predictive diagnostics.

**Data availability statement:** The dataset for this study was curated from reputable genomic databases*, including the National Center for Biotechnology Information (NCBI). The dataset comprises DNA sequences associated with Diabetes Mellitus, specifically Type 2 Diabetes Mellitus (T2DM), as well as sequences from non-diabetic individuals to serve as control samples. The sequences were obtained in FASTA format, a common format for storing nucleotide sequences. The dataset includes a total of 1,296 sequences for diabetic cases and an equal number of non-diabetic sequences, ensuring a balanced dataset for initial analysis.
* https://www.kaggle.com/datasets/fthnaja/diabetic-human-dna

**Funding:** The author(s) received no specific funding for this work.

**Competing interests:** The authors have declared that no competing interests exist.

## 1. Introduction

Diabetes Diabetes Mellitus, commonly referred to as diabetes, is a chronic metabolic disorder characterized by elevated levels of blood glucose, which, over time, can lead to serious damage to various organs and systems within the body, especially the nerves and blood vessels. According to the International Diabetes Federation, as of 2021 [1], approximately 537 million adults (20–79 years) were living with diabetes globally, and this number is projected to rise to 643 million by 2030 and 783 million by 2045. The disease is broadly categorized into Type 1, Type 2, and gestational diabetes, with Type 2 Diabetes Mellitus (T2DM) being the most prevalent, accounting for around 90% of all diabetes cases.

The increasing prevalence of diabetes, particularly Type 2, has made early diagnosis and intervention crucial. Traditionally, the diagnosis of diabetes has relied on clinical measures such as fasting blood glucose levels, HbA1c (glycated hemoglobin), and oral glucose tolerance tests [2]. While these methods are effective in diagnosing the disease, they do not address the underlying genetic factors that contribute to an individual's susceptibility to diabetes. With the advent of genomic technologies, there is growing interest in exploring the genetic basis of diabetes, particularly in identifying genetic markers that can predict an individual's risk of developing the disease.

Advances in genomics have revealed that diabetes, especially Type 2, has a significant genetic component, with multiple genes influencing an individual's susceptibility [3]. Genome-wide association studies (GWAS) have identified several loci associated with T2DM, but translating these findings into clinical practice remains challenging due to the complex interplay of multiple genetic variants and environmental factors [4,5]. Machine learning (ML) offers a powerful approach to unraveling these complex relationships by enabling the analysis of large-scale genomic data to identify patterns and associations that might not be evident through traditional statistical methods [6,7].

In the past few years, machine learning models have proved themselves as useful in the genomics field, specifically the DNA sequence classification. The models being used widely are Support Vector Machines (SVM), including NuSVC (Nu-Support Vector Classification), and ensemble methods such as XGBoost (Extreme Gradient Boosting). They maintain the ability to manage data that is high-dimensional and complex associations amongst features [8,9]. If the Natural Language Processing (NLP) techniques are integrated, the models can be enhanced further. These techniques consider DNA sequences as textual data and help manage sophisticated feature extraction procedures like TF-IDF (Term Frequency-Inverse Document Frequency) [10].

Even though the ML techniques have advanced, further development is required to apply the models to DNA sequence classification, specifically for the prediction of diabetes susceptibility. The research objective is to fill all gaps through a comparison of NuSVC and XGBoost performance when classifying the DNA sequences related to Diabetes Mellitus. The database used for the research is based on reputable sources, along with the National Center for Biotechnology Information (NCBI). The

performance of the model can be enhanced through the application of advanced preprocessing and feature extraction techniques.

Machine learning being applied in genomics is assessed in the current research. DNA sequence classification for Diabetes Mellitus is the specific context being applied. Initially, a comparative framework would be introduced, which would systematically assess the two advanced models, NuSVC and XGBoost, for the classification of DNA sequences. Furthermore, the Natural Language Processing (NLP) techniques are integrated within the research by using DNA sequences as textual data and the application of TF-IDF transformation. The techniques and their potential to enhance genomic data analysis can be observed. Also, a common genomic research issue mentioned in the research is class imbalance. The Synthetic Minority Over-Sampling Technique (SMOTE) has been applied to carry out model performance that is accurate and balanced.

The outcomes of the research present practical insights regarding the model application within clinical settings. Superior performance was indicated by XGBoost throughout the evaluated metrics. This tool can be useful in attaining early diagnosis and diabetes management, considering genetic markers. The genomic data analysis field is contributed to by research but can influence clinical practices through the provision of accurate tools for the assessment of genetic risk. The following is the structure of the remaining paper. The methodology is outlined in Section 2 – it includes data collection, preprocessing, and machine learning models that use a detailed description. Section 3 mentions the outcomes, including the key metrics like precision, accuracy, recall, log loss, ROC-AUC and F1-score. Furthermore, the research implications would be discussed where the NuSVC and XGBoost performance would be compared and possible applications within clinical practice. Lastly, the paper is concluded in Section 4 by summarizing the important findings and indicating the future research directions.

## 2. Related work

The application of machine learning (ML) in genomics has advanced the understanding and prediction of complex diseases such as Diabetes Mellitus (DM). ML models offer the capability to extract hidden patterns from high-dimensional biological data, aiding in the classification of genetic sequences and identification of disease-associated variants. The evolution of ML in this domain can be categorized into three major streams: classical models for feature selection, ensemble methods for complex relationships, and deep learning for raw sequence modeling.

### 2.1 Classical machine learning models and feature selection in genomics

For genomic classification, the earliest model developed was the Support Vector Machines (SVMs) since they can manage high-dimensional and sparse data (Huang et al. [11] and Guyon et al. [12]). SVMs are quite effective in carrying out gene selection tasks over microarray data, specifically for the classification of cancer. The approach has been extended towards the genomic sequence classification, validating the ability of the model to detect the genetic patterns that are class specific. The foundational research presented the basis to apply linear and kernel-based classifiers to the biological data.

### 2.2 Ensemble learning for predictive modeling of genetic risk

Complexities in the models have increased, and since then, ensemble techniques, specifically, Random Forests (RF) and Extreme Gradient Boosting (XGBoost), have attracted attention due to their superior interpretability and predictive performance. XGBoost has been regarded as a scalable ensemble algorithm by Chen and Guestrin [9] and was then applied to genome-wide association studies (GWAS) since it increased feature attribution and phenotypic trait prediction (Lundberg et al. [13]).

López et al. [14] employed RF on SNP data from 677 Spanish individuals, reporting an AUC of 0.89, which outperformed SVM and logistic regression. In a large-scale study, Huang et al. [15] combined SNP-based polygenic risk scores and imaging data using XGBoost across 68,911 individuals in the Taiwan Biobank, achieving an AUC of 0.94. Similarly, Hahn et al. [16] used RF in the KoGES cohort with 239,062 SNPs and metabolomic profiles, reaching an AUC of 0.876. These findings support the robustness of ensemble models in handling genotype–phenotype relationships and multi-modal inputs.

## 2.3 Deep learning approaches for DNA and SNP sequence classification

With the increasing availability of raw genetic sequence data, deep learning models have been applied to directly model nucleotide patterns. El-Attar et al. [17] designed a CNN-LSTM hybrid network trained on 300 insulin gene sequences from GenBank, achieving 99% classification accuracy, outperforming individual CNN and LSTM models. Kaur et al. [18] advanced this approach by extracting one-hot encoded 4-mer features from GenBank DNA sequences and comparing LSTM, CNN, and RF classifiers, with LSTM showing the highest performance. Kim et al. [19] demonstrated that deep neural networks (DNNs), when trained on up to 678 SNPs from two large U.S. cohorts, outperformed logistic regression and clinical models in predicting type 2 diabetes, with DNN showing consistently higher AUC values, especially when incorporating both genetic and clinical features. Srinivasu et al. [20] explored recurrent neural networks (RNNs) for classifying microRNA sequences from 10,094 entries, demonstrating potential despite constraints posed by limited sample size. These studies confirm the suitability of deep neural networks for capturing long-range dependencies and structural motifs in genomic data.

## 2.4 Natural language processing techniques and data imbalance solutions

As part of parallel developments, the Natural Language Processing (NLP) methods should be applied so that structured features can be extracted from the genomic sequences. DNA is treated like a symbolic language, and due to this, TF-IDF, k-mer frequency encoding and embedding techniques helped convert the raw sequences to meaningful input vectors. Deep learning was applied, combined with NLP-based features, so that regulatory DNA elements can be classified. They would be able to outperform the traditionally used statistical descriptors. For genomic studies, class imbalance management is quite an issue. The Synthetic Minority Over-sampling Technique (SMOTE) has been proposed by Chawla et al. [21], where synthetic samples are generated to increase the minority class representation. Since then, the technique has been applied successfully to genomic classification tasks [22], increasing model sensitivity and avoiding substantial bias.

## 2.5 Current gaps and study contribution

Limitations are there, though there are various advancements. Firstly, standardization is lacking throughout the datasets and model generalizability is restricted due to the types of diseases. Second, ensemble procedures and deep learning may be promising, there are scarce comparative assessments across architectures applying consistent genomic input formats. Thirdly, for modeling of biological sequences, the NLP-based encoding is quite effective, but there is limited complete integration within the end-to-end ML pipelines for the classification of the disease. Lastly, class imbalance continues to affect performance in datasets with underrepresented disease variants.

To address these challenges, the present study proposes a comparative analysis of NuSVC and XGBoost classifiers for DNA-based diabetes prediction. The approach integrates k-mer NLP encoding and applies SMOTE for balancing the dataset. This framework contributes to the growing body of work exploring interpretable, accurate, and generalizable models for genomic disease classification.

## 3. Methodology

The methodology of this study is structured into five main phases: data collection, preprocessing, feature extraction, model training, and performance evaluation, as illustrated in Fig 1. DNA sequence data related to diabetic and non-diabetic individuals were sourced from trusted genomic repositories.

### 3.1 Data collection

The dataset for this study was curated from reputable genomic databases, including the National Center for Biotechnology Information (NCBI). The dataset comprises DNA sequences associated with Diabetes Mellitus, specifically Type 2 Diabetes

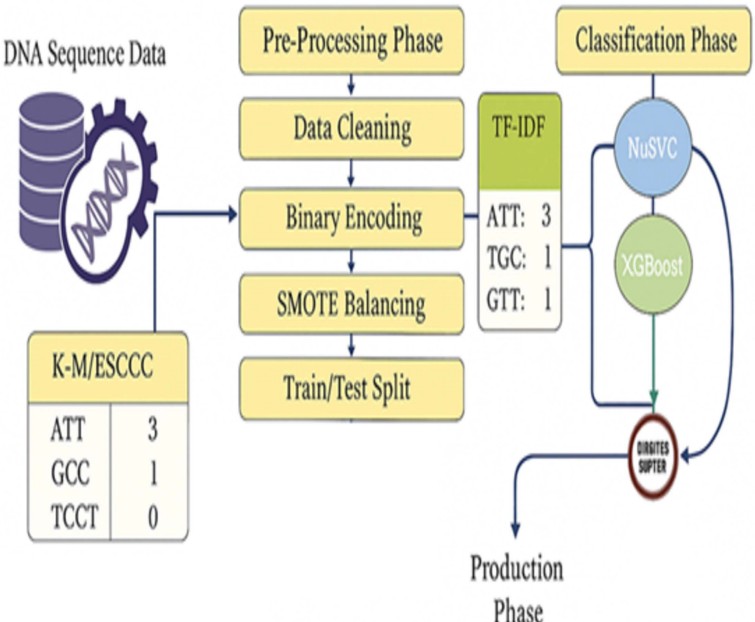

**Fig 1. Workflow of DNA-based diabetes prediction using NuSVC and XGBoost.**

Mellitus (T2DM), as well as sequences from non-diabetic individuals to serve as control samples. The sequences were obtained in FASTA format, a common format for storing nucleotide sequences. The dataset includes a total of 1,296 sequences for diabetic cases and an equal number of non-diabetic sequences, ensuring a balanced dataset for initial analysis [23].

The selection of sequences was guided by the presence of genetic markers identified in previous genome-wide association studies (GWAS) as being linked to T2DM. These markers include single-nucleotide polymorphisms (SNPs) and other variants that have been implicated in the disease. By focusing on these markers, the study aims to enhance the specificity of the classification models.

### 3.2 Data preprocessing

To make sure the raw DNA sequences were structured, clean and suitable for classification, implementation of a comprehensive preprocessing pipeline was implemented. Firstly, the BioPython library was used to parse the DNA sequences from two FASTA files (diabetic and non-diabetic). To maintain data integrity, sequences which were ambiguous or with nucleotide characters had been removed. The sequence was then labelled with a binary class (1 for diabetic, 0 for non-diabetic). Discriminative variables were extracted by applying the TF-IDF vectorization through 3–6-character n-grams. Each DNA sequence was effectively treated as a string of overlapping nucleotide tokens. Hence, the machine learning models learnt from expressive sequence motifs. Considering the class imbalance, the synthetic minority class examples and the dataset were generated through the Synthetic Minority Over-sampling Technique (SMOTE). Lastly, training and test sets were made using the dataset by applying the 80:20 stratified approach so that class proportions could be preserved. The manuscript now integrates model training stages, resampling, feature extraction and data cleaning so that transparency and clarity can be enhanced.

### 3.3 Feature extraction

DNA sequences are converted to a suitable format for machine learning through character-level feature extraction, which is Term Frequency–Inverse Document Frequency (TF-IDF). Each of the sequences was considered a text string, and the

TfidfVectorizer was used for processing with the analyzer='char' and ngram_range= (3,6). This generated an overlapping k-mers of lengths 3–6. With the help of this range, short and moderately long nucleotide patterns could be captured, which reflect the biologically expressive motifs.

Let each DNA sequence be treated as a document $d$, composed of overlapping k-mers $t_i$. The TF-IDF score for a given k-mer $t_i$ in sequence $d$ is calculated as [24]:

$$TF - IDF\ (t_i, d) = tf\ (t_i, d) \cdot \log \left( \frac{N}{df(t_i)} \right)$$

(1)

where $tf\ (t_i, d)$ is the frequency of k-mer $t_i$ in document $d$, $N$ is the total number of sequences, and $df)\ t_i$ (is the number of sequences containing $t_i$.

The TF-IDF matrix helped encode the frequency-weighted presence of the k-mers throughout the sequences, which helped produce a high-dimensional, sparse feature set. Input X is the matrix served, and the corresponding labels (y) were the classification NumPy array. Through this method, the NuSVC and XGBoost models effectively learnt from diabetes related structural composition of DNA sequences.

## 3.4 Handling class imbalance

Within genomic research, class imbalance is a vital issue, specifically when disease-specific data sets are being used in which genetic markers are rare. Within the current research, any training set imbalance was addressed through the Synthetic Minority Over-sampling Technique (SMOTE). Synthetic samples are generated by SMOTE by interpolating amongst the current minority class samples, which effectively enhances the minority class representation by duplicating present data. With this approach, model bias is prevented regarding the majority class and enhances the model's generalization towards unseen data.

To address class imbalance, SMOTE generates synthetic data points using [21]:

$$x_{new} = x + \delta \cdot (x_{nn} - x) , \ \delta \in [0, 1]$$

(2)

where $x$ is a minority class sample, $x_{nn}$ is a randomly selected nearest neighbor, and $\delta$ is a random number between 0 and 1.

NuSVC solves the following optimization problem [25]:

$$\min_{w,b,\xi} \ \frac{1}{2} \|w\|^2 + \nu C \sum_{i=1}^{n} \xi_i$$

(3)

$$\text{subject to } y_i \left( w \cdot \phi (x_i) + b \right) \geq 1 - \xi_i, \ \xi_i \geq 0$$

where $\phi (x_i)$ is the kernel-transformed input, $\xi_i$ is the slack variable, and $\nu \in (0, 1]$ controls the trade-off. XGBoost minimizes a regularized objective function [9]:

$$\mathcal{L} = \sum_{i=1}^{n} l(y_i, \hat{y}_i) + \sum_{k=1}^{K} \Omega (f_k)$$

(4)

$$\Omega(f) = \gamma T + \frac{1}{2} \lambda \sum_{j=1}^{T} w_j^2$$

(5)

where $l$ is the log loss, $f_k$ is the kth tree, $T$ is the number of leaves, and $w_j$ is the score on leaf $j$. The regularization terms $\gamma$ and $\lambda$ control complexity to prevent overfitting.

## 3.5 Model training

DNA Sequences were classified using NuSVC (Nu-Support Vector Classification) and XGBoost (Extreme Gradient Boosting), which are advanced machine learning models. The models are known for their effectiveness in managing complex feature associations and high-dimensional data.

- **NuSVC**: NuSVC is a traditional Support Vector Machine (SVM)variant which includes the parameter 'nu' so that the margin errors and support vectors could be controlled. The model is particularly useful for cases in which there is no linear separation of data. It can manage non-linear associations within data by applying kernel functions. Within the current research, the radial basis function (RBF) kernel has been applied, and grid search was used to optimize the hyperparameters so that maximum model performance could be attained.

- **XGBoost**: XGBoost uses gradient boosting, which is an ensemble learning method. A decision tree series is developed where the tree is focused on correcting errors that were created earlier. There is scalability and high performance of XGBoost, which is why it is considered the best choice for genomic data as it is complex and large. For optimal performance, cross validation was done to tune hyperparameters like boosting rounds numbers, maximum tree depth and learning rate.

The model-specific configurations were used for the two models for training of the pre-processed and feature-extracted dataset. There were 300 boosting rounds used for the training of XGBoost, and a 10-fold cross-validation was applied for the NuSVC. The parameters were optimized in each model so that classification errors could be minimized considering the appropriate objective function.

## 3.6 Model evaluation

The validation set was applied for model evaluation, and performance was assessed by applying various metrics.

- **Accuracy**: For each model, overall accuracy was calculated as the proportion of correctly classified sequences out of the total number of sequences.

- **Precision, Recall, and F1-Score**: For each class (diabetic and non-diabetic), the metrics were calculated so that model performance could be understood clearly. Amongst the predicted positives, the precision measures the true positive proportion, recalls the measures of the true positive proportions within actual positives, and the harmonic mean is the F1-score of the precision and recall.

- **ROC-AUC**: To assess the trade-off between the false positive and true positive rates, the calculation of the Receiver Operating Characteristic (ROC) curve and the corresponding Area Under the Curve (AUC) score was done. If the AUC is high, the model performs better in stating the difference between diabetic and non-diabetic sequences.

- **Log Loss**: Model performance measurements were carried out using Log Loss considering the provided probability estimates. When the log loss is lower, the model calibration is better, since the predicted probabilities are close to actual class blends.

- **Confusion Matrix**: Performance classification, in terms of true positive, true negative, false positive, and false negative predictions for both models was visualized through confusion matrices.

The outcomes of the assessments have been compared to indicate each model's strengths and weaknesses. XGBoost indicated superior performance throughout the metrics and was robust in managing complex genomic data. For the SVM-based approach's effectiveness, valuable insights were provided by NuSVC.

## 4. Results

The test dataset included 519 DNA sequences, which were used for the evaluation of the NuSVC and XGBoost model performance. For each model, the key evaluation metrics are log loss, ROC-AUC, a confusion matrix, F1-score, recall, precision and accuracy. According to the research outcomes, the two models have quite a difference in performance. Throughout the metrics, the NuSVC was outperformed by the XGBoost in a significant manner.

### 4.1 Classification results

Classification performance can be evaluated by accuracy. Within the current research, NuSVC has an accuracy of 86.90%, and XGBoost has 97.88%. The 11% difference shows improvement for XGBoost and its superior ability to appropriately classify the DNA sequences associated with diabetes susceptibility. The difference would be considered reliable after pairing a t-test throughout the 10-fold cross-validation folds. It has been confirmed through results that XGBoost consisted of accuracy enhancement and was statistically significant ($p < 0.05$), which shows its ability to manage complex genomic data that is complex.

The classification reports for both models are summarized below:

- **NuSVC Classification Report**

    - Precision: 0.79 (non-diabetic), 0.99 (diabetic)

    - Recall: 1.00 (non-diabetic), 0.74 (diabetic)

    - F1-Score: 0.88 (non-diabetic), 0.85 (diabetic)

    - Overall Accuracy: 86.90%

- **XGBoost Classification Report**

    - Precision: 0.97 (non-diabetic), 0.99 (diabetic)

    - Recall: 0.99 (non-diabetic), 0.97 (diabetic)

    - F1-Score: 0.98 (both classes)

    - Overall Accuracy: 97.88%

Fig 2 presents the Receiver Operating Characteristic (ROC) curve, which indicates a trade-off amongst the two models' true positive rate (sensitivity) and false positive rate (1-specificity). The summary measure is mentioned as part of the Area Under the Curve (AUC), which indicates the ability of the model to differentiate among the classes. The AUC of XGBoost is 1.00, and this shows that it is a perfect model. The AUC of NuSVC is 0.95. According to the ROC curve, the NuSVC has been outperformed consistently by the XGBoost throughout all thresholds. Hence, there is clear evidence that XGBoost is robust in managing classification activities.

The probability estimate accuracy of the models is measured by Log loss, which is also referred to as cross-entropy loss or logistic loss. Performance is better if the log loss is lower, where the actual labels and predicted probabilities should be close. It is shown in Fig 3 that the log loss of NuSVC was 0.2649, which is higher than the log loss of XGBoost, which is 0.0650. It is through this difference that one can note the superior calibration of XGBoost towards probability estimates. It has helped make the real-world applications quite reliable and considers the probability-based decisions to be vital.

Fig 4 shows the training and validation loss trends during model evaluation. For XGBoost, loss remained consistently low across boosting rounds, indicating stable convergence and strong generalization. In contrast, NuSVC displayed greater variability in loss across cross-validation folds, suggesting potential overfitting. The overall validation loss for

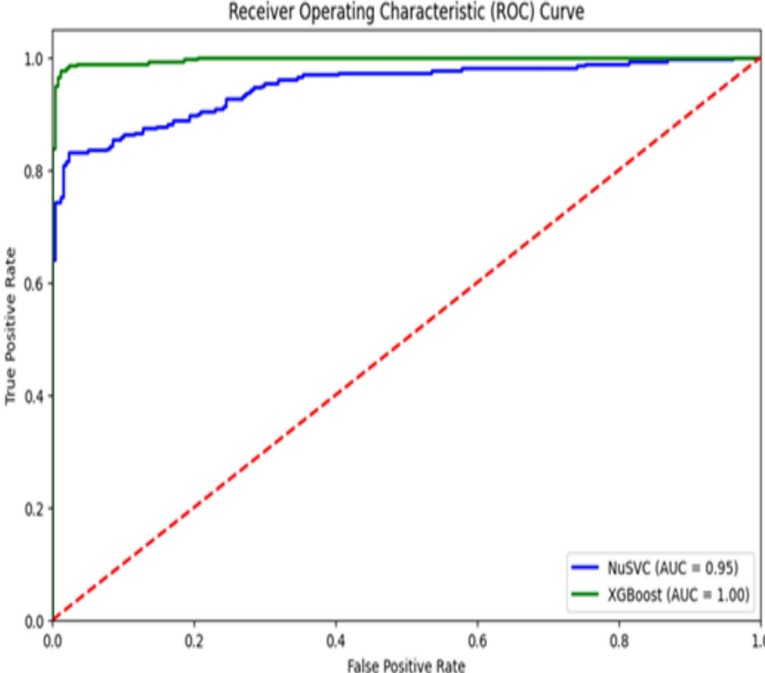

**Fig 2. ROC curve.**

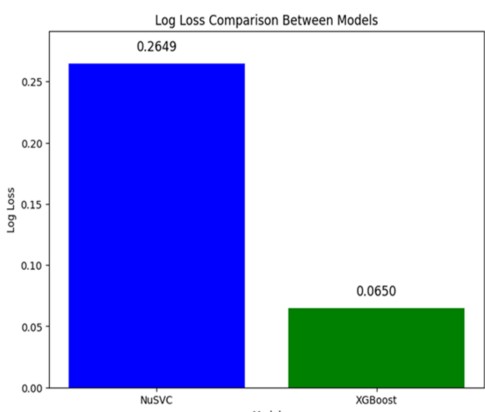

**Fig 3. Log loss.**

XGBoost was consistently lower than that of NuSVC, reinforcing its superior generalization performance on the genomic dataset.

The confusion matrices for NuSVC and XGBoost are presented in Fig 5 and Fig 6, respectively. These matrices provide detailed insights into the classification performance of each model:

- NuSVC Confusion Matrix (Fig 5): The NuSVC model correctly classified 257 of the 258 non-diabetic sequences (class 0) and 194 of the 261 diabetic sequences (class 1). However, it misclassified 67 diabetic sequences as non-diabetic, leading to a lower recall for the diabetic class.

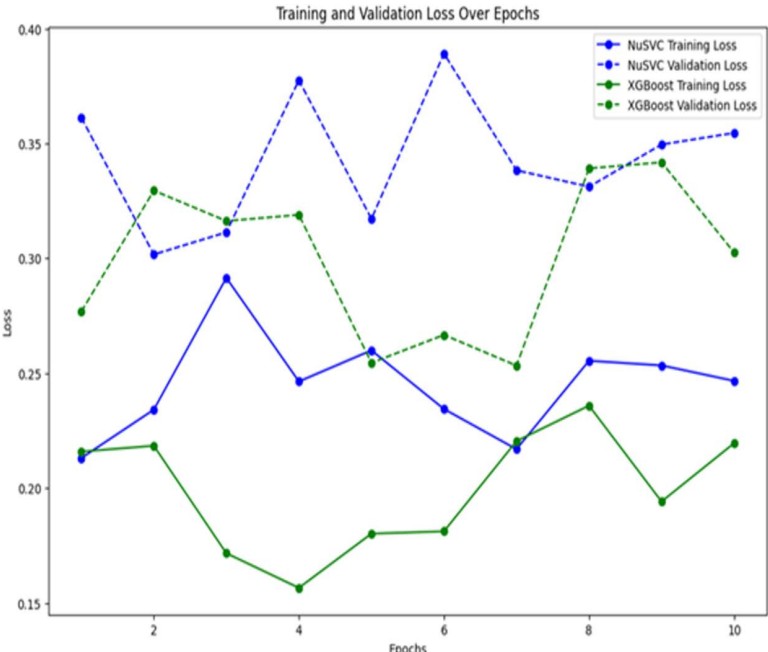

**Fig 4. Training and validation loss.**

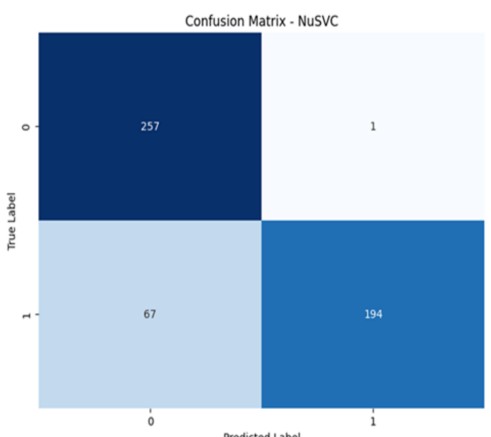

**Fig 5. NuSVC confusion matrix.**

- Confusion Matrix (Fig 6): Superior performance was achieved by the XGBoost model. Out of 258 non-diabetic sequences, 256 were correctly classified, and out of 261 diabetic sequences, 252 were correctly classified. A few sequences were only misclassified, 2 non-diabetic as diabetic and 9 diabetic as non-diabetic. This shows that they were highly accurate and maintained balanced performance throughout the classes.

    The results show a clear performance gap between XGBoost and NuSVC. XGBoost achieved higher precision, recall, F1-score, and AUC, suggesting better generalization and robustness on genomic data. Its AUC of 1.00 and log loss of 0.0650 confirm that the model not only classifies well but also assigns accurate probabilities—important for medical

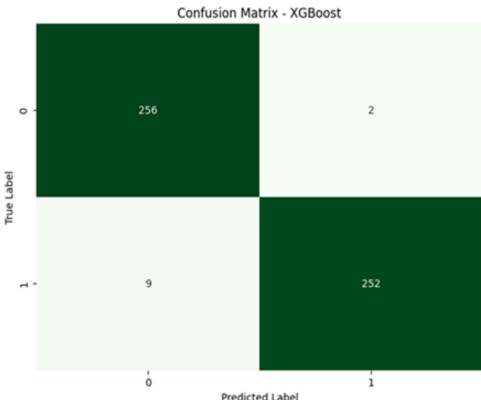

**Fig 6. XGBoost confusion matrix.**

decision-making. NuSVC struggled with recall for the diabetic class (0.74), leading to more false negatives. This suggests that NuSVC was less sensitive to detecting diabetic cases, which poses a clinical risk if used in real diagnosis. The confusion matrix supports this, showing 67 diabetic samples misclassified as non-diabetic. In contrast, XGBoost misclassified only 9 diabetic cases.

The overfitting pattern in NuSVC is also evident in the loss curve (Fig 3), with training loss improving while validation loss fluctuates. This indicates NuSVC learned noise in the training data, reducing its ability to generalize. XGBoost maintained stable loss throughout, showing better regularization and convergence.

SMOTE improved class balance, but XGBoost was more effective in leveraging the synthetic samples. NuSVC, though optimized via grid search, lacked adaptive learning capacity and may have required further kernel tuning.

Overall, XGBoost showed superior performance by effectively handling imbalanced and high-dimensional feature spaces, producing well-calibrated probability estimates, and maintaining high sensitivity in detecting diabetic cases. This is crucial in clinical applications, where minimizing false negatives directly impacts patient safety. The results suggest that XGBoost offers a more reliable and safer approach for DNA-based diabetes classification compared to NuSVC.

## 5. Discussion

The comparative results between NuSVC and XGBoost show a clear and consistent advantage for XGBoost in classifying DNA sequences associated with Type 2 Diabetes Mellitus. XGBoost achieved superior scores across all performance metrics, including accuracy (97.88%), AUC (1.00), and F1-score (0.98), demonstrating its robustness in handling genomic data transformed via TF-IDF. In contrast, NuSVC underperformed in recall for diabetic cases (0.74), a critical metric in clinical screening tasks where false negatives could lead to undiagnosed patients and delayed intervention. Similar findings were reported by Chen and Guestrin [9], who emphasized XGBoost's strength in handling structured and sparse datasets with high accuracy and generalizability.

XGBoost's advantage is rooted in its ability to model complex, non-linear feature interactions using additive tree-based models with gradient boosting. It also demonstrated strong probability calibration, with a log loss of 0.0650, indicating reliable prediction confidence levels. Studies by Lundberg et al. [13] support this, showing that XGBoost not only achieves high classification performance but also provides interpretable feature attribution for biomedical data. The model's performance curves showed minimal overfitting across 10 training epochs, unlike NuSVC, which displayed greater variance in training and validation loss. Even with SMOTE applied, NuSVC showed signs of model instability, likely due to its sensitivity to kernel parameters and its limited flexibility in learning from high-dimensional sparse vectors.

In clinical prediction contexts, models must prioritize sensitivity, especially for identifying diabetic patients. XGBoost maintained a recall of 0.97 for diabetic cases and misclassified only 9 diabetic samples, versus 67 for NuSVC. Missing true positives in diabetes prediction can lead to serious downstream health complications, such as retinopathy, nephropathy, and cardiovascular disease, as shown in cohort analyses by Gaulton et al. [5].

From a computational efficiency perspective, XGBoost also scales better with TF-IDF feature representations derived from k-mer tokenization. TF-IDF matrices are sparse and high-dimensional, which poses challenges for kernel-based models like NuSVC. As noted by Müller and Guido [26], linear models or tree ensembles tend to outperform kernel-based models in such scenarios, especially when regularization is embedded into the model structure.

Our pipeline—based on k-mer segmentation, TF-IDF transformation, and ensemble classification—is simple to reproduce, transparent for interpretation, and adaptable for different genomic diseases. Unlike complex black-box models, this framework provides both performance and explainability. However, deep learning techniques (Transformer) offer additional potential. Recent studies, such as Permatasari et al. [27], applied attention-based SVM models to similar genomic tasks, achieving slightly lower accuracy than our XGBoost results, but with higher computational complexity. Soo et al. [28] also showed that transformer-based models can effectively integrate multi-omics data for diabetes prediction, though they require more computational resources and larger datasets for training. The proposed deep learning model outperformed traditional methods (RF, SVM) and EI-DNN in classifying specific Type 1 Diabetes (T1D) complications, achieving peak accuracies such as 0.9038 for neurological and 0.9410 for ophthalmic complications, along with a recall of 0.9327 for other complications. However, the model showed limited improvement in predicting ketoacidosis and struggled with multiple or unspecified complications, reflecting challenges posed by data sparsity and class imbalance. A comprehensive comparison of our model and these recent approaches is provided in Table 1, highlighting the trade-offs between accuracy, interpretability, and resource requirements.

Future research should focus on incorporating deep architectures such as transformers or hybrid CNN-RNNs to capture richer spatial and temporal dependencies in genetic data. Additionally, generalization could be improved by testing across multi-ethnic or multi-condition datasets and validating the model in clinical cohorts. By balancing performance, simplicity, and real-world relevance, this study contributes a reproducible benchmark that can support scalable, early-stage genomic disease screening.

## 6. Conclusion

Diabetes Mellitus, specifically Type 2 Diabetes (T2DM), is considered to be a global health issue. It is necessary to attain a diagnosis at an early stage and implement management strategies accordingly. The genetics of diabetes has been thoroughly understood, and it is observed that genetic data should be classified and disease susceptibility predicted using machine learning models. The DNA sequences associated with T2DM were classified by applying NuSVC (Nu-Support

Table 1. Comparison of genomic diabetes prediction models.

| Study | Model Type | Feature Representation | Accuracy | AUC | Key Strengths | Limitations |
|---|---|---|---|---|---|---|
| **This Study** (2025) | TF-IDF (k-mer) + XGBoost vs. NuSVC | TF-IDF on 3-mer and 6-mer | 97.88% | 1.00 | High performance, interpretable, fast | Shallow model, no raw embedding use |
| Permatasari et al. (2025) [27] | SVM+Transformer | Entropy-based mapping | 99.89% | Not reported | Transformer module for deep feature extraction | Requires complex pre-processing (entropy, STFT, imaging) |
| Soo et al. (2025) [28] | Dual DNN+Transfer Learning | Indels and SNPs from UK Biobank (T2D→T1D) | 94.00% | Not reported | First to use TL for T1D complications; improved performance in key categories | Lower performance for sparse classes; limited ethnic diversity in data |

Vector Classification) and XGBoost (Extreme Gradient Boosting). These are machine learning models that are quite advanced. The current research objective is asses and compare the model performance so that the DNA sequences can be classified accurately. In this manner, the broad objective would be to develop predictive diagnostics within genomics. The research results show that the two models have quite a difference in performance levels. Throughout the assessed metrics, the NuSVC was outperformed by the XGBoost. The NuSVC had an accuracy level of 86.90%, and XGBoost's was 97.88%. For ROC-AUC, superior performance was indicated by XGBoost with 1.00 AUC. It shows that their classification abilities are perfect. On the other hand, NuSVC has a 0.95 AUC. Furthermore, the log loss for XGBoost was 0.0650, which is significantly lower than compared of NuSVC, which is 0.2649. This indicates that their model reliability and calibration are much better. The XGBoost also has fewer misclassifications according to the confusion matrix analysis. This is specifically for diabetic sequences. Hence, the accuracy and robustness are reinforced. Through these outcomes, one can recognize XGBoost as an effective model to carry out DNA sequence classification regarding diabetes risk prediction. For personalized healthcare and genomic medicine, the research implications are quite significant. Considering genetic markers, T2DM can be diagnosed early using the XGBoost since it maintains superior performance and can be integrated effectively within clinical workflows. Individuals who are at a high risk of getting diabetes can be identified accurately, personalized intervention strategies can be implemented by healthcare providers, which would possibly reduce the disease severity and incidence. A methodological framework is brought forward by using the feature extraction NLP techniques and class imbalance-related SMOTE. This framework can then be applied to the rest of the genomic research, possibly enhancing the machine learning model's reliability and accuracy throughout the genetic classification tasks. Even though the outcomes are quite promising, the research faces various limitations that need to be considered. First, even though the dataset has been extracted from reliable sources, it does not integrate the genetic variant diversity related to T2DM throughout the populations. The research generalizability could be limited and not include diverse and broad populations. Second, two advanced models have only been integrated, it is possible that other deep learning models or machine learning would present performance advantages or insights which were not assessed as part of this research. Last, the research relies upon particular hyperparameters to carry out model training. Hence, it is possible that the outcomes would vary across the various parameter settings, possibly influencing the comparative performance of the model. The limitations can be addressed in future research being conducted through data expansion and inculcating a diverse genetic sequence range throughout several populations. This would help generalize the research outcomes. Moreover, research models and deep learning architectures like Recurrent Neural Networks (RNNs) or Convolutional Neural Networks (CNNs) would also prove to be useful in providing insights for effective DNA sequence classification approaches. The predictive power of the models can be enhanced by including multi-omics data and integrating genomics with metabolomics, transcriptomics and proteomics. It would also present a holistic environmental variables and genetic factors which contribute towards T2DM. To conclude, the continuing research must validate and develop models within a clinical setting, making sure they are reliable, robust and can be integrated within routine healthcare practices. The current research highlights the importance of machine learning, specifically XGBoost, for the advancement of DNA sequence classification and early diabetes diagnosis. The limitations need to be addressed and findings developed so that the models can be further refined in the future. The healthcare solutions should be data-driven and personalized so that they may be able to fight chronic diseases such as T2DM.

## Author contributions

**Conceptualization:** Said A. Salloum, Khaled Mohammad Alomari.

**Formal analysis:** Khaled Mohammad Alomari.

**Methodology:** Khaled Mohammad Alomari.

**Project administration:** Said A. Salloum.

**Supervision:** Said A. Salloum.

**Validation:** Ayham Salloum.

**Visualization:** Ayham Salloum.

**Writing – original draft:** Said A. Salloum.

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
