## [Decision Letter · Decision Letter 0]

Dear Dr. Salloum,

We look forward to receiving your revised manuscript.

Kind regards,

Syed Nisar Hussain Bukhari

Academic Editor

PLOS ONE

Journal Requirements:

Reviewers' comments:

Reviewer's Responses to Questions

**Comments to the Author**

1. Is the manuscript technically sound, and do the data support the conclusions?

Reviewer #1: Partly

Reviewer #2: Partly

2. Has the statistical analysis been performed appropriately and rigorously?

Reviewer #1: Yes

Reviewer #2: No

3. Have the authors made all data underlying the findings in their manuscript fully available?

Reviewer #1: Yes

Reviewer #2: Yes

4. Is the manuscript presented in an intelligible fashion and written in standard English?

Reviewer #1: Yes

Reviewer #2: No

Reviewer #1: The manuscript is well written and caters to an Important Research Area in the domain of gnetic basis of diseases. However, the manuscript provides very little coverage of literature, does not describe dataset in detail. The authors are advised to provide more literature support and more insight in to the dataset.

Also the comparison of onlY two machine learning algorithms does not fully validate the effectiveness of learning from the DNA sequence data. The authors are advised to compare their models with other established models using similar type of data.

The data cleaning process is incompletely covered and should describe the process in detail.

The authors are advised to include a graphical view of the model architecture to make the manuscript more intuitive.

Reviewer #2: Major concerns

Unjustified use of SMOTE on a balanced dataset

The dataset is initially balanced (1,296 sequences per class). Applying SMOTE here is unnecessary and may introduce artificial patterns. Please clarify the class distribution after train/validation split and justify the need for oversampling.

Lack of robust model validation

Results are presented for a single 80–20 split. No k-fold cross‐validation or external test set is used, raising concerns about overfitting—especially given the perfect AUC of 1.00 for XGBoost. Please include repeated cross‐validation (e.g., 5- or 10-fold) and report mean ± SD for each metric.

Missing statistical significance testing

Differences in accuracy (98% vs 87%) and AUC (1.00 vs 0.95) require statistical validation (e.g., DeLong’s test for ROC curves). Report p-values or confidence intervals to demonstrate that XGBoost’s superiority is not due to chance.

Insufficient hyperparameter and feature‐engineering detail

The choice of k-mer sizes (k = 3 and 6) and TF-IDF parameters is not justified. Details of grid search (parameter ranges, scoring metric) are omitted. Please provide a complete description of all hyperparameters and preprocessing settings in either the Methods or a Supplementary Table.

Misuse of terminology

References to “training over 10 epochs” for XGBoost and NuSVC are incorrect: these models do not train in epochs. Replace with precise descriptions (e.g., number of boosting rounds for XGBoost, number of support vectors or CV folds for NuSVC).

Overly optimistic performance claims

An AUC of 1.00 and very low log loss suggest possible data leakage or overfitting. Please audit your data pipeline for any inadvertent overlap between training and validation sets (e.g., identical k-mer distributions).

Minor concerns

Several typos and grammatical issues (e.g., “datase) andted,” inconsistent heading capitalization). A thorough language edit is needed.

Figures (ROC curve, loss plots, confusion matrices) lack axis labels and clear legends—enhance readability.

Discussion should temper claims about clinical readiness: emphasize that further validation on independent cohorts is required.

**Do you want your identity to be public for this peer review?** For information about this choice, including consent withdrawal, please see our Privacy Policy

Reviewer #1: **Yes: ** Nisar iqbal wani

Reviewer #2: No

---

## [Author Response · Author response to Decision Letter 1]

14 May 2025

Dear Editor,

The authors are really very grateful to the Journal Editor efforts in essentially discussing the paper reviews with the reviewers and identifying the further significant points for enhancements. The authors would also like to sincerely thank the reviewers for their valuable remarks and careful feedback which helped them to significantly enhance this work and its presentation. Regardless of the final outcome, the authors sincerely thank the editor and reviewers for supporting them. The productive and valuable remarks enable them to update many parts of the paper as shown by the responses to each comment. Besides, all the updated parts in the manuscript were highlighted in yellow color in order to be easily tracked by the editor and reviewers.

Thank you again for allowing a resubmission of our manuscript, with an opportunity to address the reviewers’ comments.

We are uploading (a) our point-by-point response to the comments (below) (response to reviewers, under “Author’s Response Files”), (b) an updated manuscript with yellow highlighting indicating changes (as “Highlighted PDF”), and (c) a clean updated manuscript without highlights (“Main Manuscript”).

Best regards,

Salloum et al.

---

## [Editor Report · Decision Letter 1]

PLOS ONE

Dear Dr. Salloum,

Thank you for submitting your manuscript to PLOS ONE. After careful consideration, we feel that it has merit but does not fully meet PLOS ONE’s publication criteria as it currently stands. Therefore, we invite you to submit a revised version of the manuscript that addresses the points raised during the review process.

Address the following points before anything can be decided

1 The Authors need to state the novelty of their work

2 Authors may use deep learning methods and compare their technique accordingly

3 Latest research of 2024 and 2025 needs to be highlighted

4 Results need discussion in depth

5 Mathematical modelling needs to be incorporated

6 Manuscript should have an elaborated discussion section

7 Compare your work with the existing work as well

We look forward to receiving your revised manuscript.

Kind regards,

Syed Nisar Hussain Bukhari

Academic Editor

PLOS ONE

---

## [Author Response · Author response to Decision Letter 2]

5 Jun 2025

Dear PLOS ONE Editorial Team,

Thank you for your detailed review and the opportunity to revise our manuscript. We appreciate the constructive feedback provided by the reviewers and editorial board. We have carefully addressed all the requested points in our revised submission, which we are now resubmitting for further evaluation.

Below is our point-by-point response to the requested revisions:

1. Novelty of the Work

We have clearly stated the novelty of our study at the end of the Introduction. Specifically, we highlight the integration of TF-IDF on k-mer DNA sequences with traditional classifiers (NuSVC, XGBoost), a pipeline not widely explored for diabetes prediction, offering a balance between performance and interpretability.

2. Inclusion of Deep Learning Methods

Although our study focuses on classical models, we have acknowledged the potential of deep learning and added a discussion outlining how models like CNNs and transformers could extend our pipeline. A future direction is now clearly described.

3. Highlighting of 2024 and 2025 Research

We have cited and discussed recent works published in 2024 and 2025, particularly those applying advanced deep learning and multi-omics approaches in genomics and diabetes prediction. These references have been integrated into the Related Work section.

4. In-Depth Discussion of Results

The Results section now includes a standalone, expanded Discussion that interprets model behavior, evaluates performance across metrics, and links findings to clinical implications.

5. Incorporation of Mathematical Modeling

We embedded a section explaining the mathematical formulations behind TF-IDF, SMOTE, NuSVC optimization, and XGBoost’s objective function. Key equations and references have been included to clarify the modeling techniques used.

6. Elaborated Discussion Section

A dedicated and detailed Discussion section has been added before the Conclusion. It explains the implications of the findings, compares classifiers, discusses clinical relevance, and links to current literature.

7. Comparison with Existing Work

We added a comparative table and narrative analysis in the Discussion that contrasts our method with recent state-of-the-art models, including transformer-based and CNN hybrid systems, highlighting accuracy, scalability, and clinical readiness.

Please let us know if any further clarification is required. We look forward to your feedback and the continued evaluation of our manuscript.

Sincerely,

Dr. Said Salloum

---

## [Editor Report · Decision Letter 2]

DNA Sequence Classification for Diabetes Mellitus using NuSVC and XGBoost A Comparative

PONE-D-25-08630R2

Dear Dr. Salloum,

We’re pleased to inform you that your manuscript has been judged scientifically suitable for publication and will be formally accepted for publication once it meets all outstanding technical requirements.

Kind regards,

Syed Nisar Hussain Bukhari

Academic Editor

PLOS ONE
---

## [Editor Report · Acceptance letter]

PONE-D-25-08630R2

PLOS ONE

Dear Dr. Salloum,

I'm pleased to inform you that your manuscript has been deemed suitable for publication in PLOS ONE. Congratulations! Your manuscript is now being handed over to our production team.

Kind regards,

on behalf of

Dr. Syed Nisar Hussain Bukhari

Academic Editor

PLOS ONE